# Associations of Atrial Fibrillation with Mild Cognitive Impairment and Dementia: An Investigation Using SPRINT Research Materials

**DOI:** 10.3390/jcm11195800

**Published:** 2022-09-30

**Authors:** Aniqa B. Alam, Ambar Kulshreshtha, Linzi Li, Vinita Subramanya, Alvaro Alonso

**Affiliations:** 1Department of Epidemiology, Rollins School of Public Health, Emory University, Atlanta, GA 30322, USA; 2Department of Family and Preventive Medicine, School of Medicine, Emory University, Atlanta, GA 30322, USA

**Keywords:** dementia, atrial fibrillation, ascertainment

## Abstract

Background: Atrial fibrillation (AF) is linked to increased risk of dementia and cognitive decline, but whether AF and its ascertainment methods affect cognition in patients with hypertension has received less attention. Methods: We studied 8469 participants with elevated systolic blood pressure who were free of stroke and diabetes at baseline enrolled in the Systolic Blood Pressure Intervention Trial. AF was ascertained using three approaches: self-report of AF, AF from a safety event, and study electrocardiogram-determined (ECG) AF. Mild cognitive impairment (MCI) and probable dementia (PD) were ascertained from in-person assessments or telephone interviews from the participant or an informant. We used Cox proportional hazard models to estimate hazard ratios for the association of AF (all three sources) with outcomes of MCI, PD, and a composite MCI/PD outcome. Results: During a mean follow-up of 4.6 years, 974 (12%) participants had AF (prevalent or incident), 634 were diagnosed with MCI, and 316 with PD. When comparing those with AF (from any source) to those without, no differences were detected in the risk of MCI or PD. Comparison between AF sources found ECG-AF to be associated with an elevated risk of MCI/PD (hazard ratio (HR) 1.59, 95% confidence interval (95%CI) 1.06, 2.38). Neither AF ascertained through safety events nor self-reported AF were associated with MCI or PD. Conclusion: The association of AF with incidence of MCI/PD differed by method of AF ascertainment. Case definition of AF and quantification of AF burden are important factors in studies evaluating the link between AF and cognitive dysfunction.

## 1. Introduction

Atrial fibrillation (AF) is one of the most common arrythmias, particularly among older persons [1]. Aging itself is the most important risk factor for AF and is associated with abnormal electrical and structural changes within the heart that may make older individuals more susceptible to developing AF [2]. By 2050, nearly a third of the entire world population is expected be over the age of 60 [2], and the prevalence of AF in the United States alone is projected to double or triple [3,4]. AF is also a risk factor for the development of dementia and cognitive decline [5]. Persons with elevated blood pressure are at particular high risk of developing both AF and cognitive dysfunction. Chronic hypertension is a highly prevalent established risk factor for AF and can compromise cerebrovascular health [6], thereby increasing the risk of dementia, particularly of the vascular type [7]. Characterizing the link of AF with dementia among hypertensives can contribute to targeted preventive strategies among this high-risk population. Therefore, using research materials from the Systolic Blood Pressure Intervention Trial (SPRINT), which includes individuals at high cardiovascular risk with elevated systolic blood pressure (BP), we explored associations of AF with mild cognitive impairment (MCI) and probable dementia (PD) over a period of 3 years. Moreover, since AF was identified from different sources, we evaluated the association of AF with MCI and PD by method of AF ascertainment, which could provide useful information to assist in the design of future studies.

## 2. Methods

The study design and results of the SPRINT and SPRINT MIND trial have been previously published [8,9]. Briefly, SPRINT was a randomized, controlled, open-label trial funded by the National Institutes of Health that compared the effects of standard hypertension treatment (target systolic BP of 140 mmHg) and intensive hypertension treatment (target systolic BP of < 120 mmHg) on four main cardiovascular endpoints: stroke, myocardial infarction, heart failure, and death due to cardiovascular causes. The SPRINT MIND ancillary study further assessed cognitive functioning in the same cohort.

### 2.1. Study Population

From November 2010 to March 2013, 9361 participants free of diabetes or prevalent stroke were recruited into the trial. Trial requirements included: being at least 50 years old, systolic BP between 130 and 180 mmHg, and an elevated risk of cardiovascular disease [9]. Exclusion from the trial was based on a number of conditions, the full list of which can be found elsewhere [8]. Those with missing cognitive diagnoses were excluded from this analysis (N = 798). After also excluding those with missing covariate information, the final analytic cohort consisted of 8469 participants (Figure 1). The study was approved by the institutional review boards at each of the field locations and participants provided written informed consent.

### 2.2. Covariates

Age in this cohort was defined at the time of treatment randomization. Race and ethnicity of participants were grouped into non-Hispanic White, non-Hispanic Black, Hispanic, and other, which comprised Native Americans, Native Hawaiians, those of Asian descent, and others. Education was categorized into three levels: less than high school, high school and vocational school, at least some college. Cardiovascular disease (CVD) history was based on self-report of angina, heart attack, or congestive heart failure at baseline. Smoking status was also based on self-report. Participants were asked to bring in current prescriptions at baseline. Oral anticoagulant (OAC) use was identified by the use of rivaroxaban, dabigatran, or warfarin. Creatinine and HDL and total cholesterol were measured in fasting blood samples at baseline. Estimated glomerular filtration rate (eGFR) was calculated from circulating creatinine. Systolic and diastolic blood pressure were the mean of three measurements performed in the same sitting at baseline. CHA₂DS₂-VASc scores were calculated to estimate the risk of stroke for AF and non-AF patients [10]. Simply, the score is the sum of risk factors known to be associated with risk of stroke in AF patients: congestive heart failure, hypertension, age ≥ 75 (counts as 2 points), diabetes, stroke (counts as 2 points), vascular disease (myocardial infarction and peripheral vascular disease), age 65–74, and female sex. Inclusion criteria required having hypertension to be part of the SPRINT trial, while also being free of diabetes at baseline (per the exclusion criteria). Thus, all participants had hypertension equal to 1 and diabetes equal to 0 when calculating CHA₂DS₂-VASc scores. Atherosclerotic Cardiovascular Disease (ASCVD) scores were also calculated from the American College of Cardiology/American Heart Association (ACC/AHA) pooled cohort equations to determine the 10-year risk of heart disease and stroke [11]. ASCVD scores are race- and sex-specific, and use age, total and HDL cholesterol, systolic blood pressure, smoking status, and diabetes status to generate risk scores. Again, as no participant had diabetes, diabetes status and its corresponding coefficients were set to 0 in the equation. Predetermined coefficients have only been generated for black and white racial/ethnic groups, so per ACC/AHA recommendations, coefficients for white non-Hispanic patients were used to calculate scores for Hispanic and other racial/ethnic groups.

### 2.3. Atrial Fibrillation

Ascertainment of AF was based on three approaches of AF diagnosis: self-reported AF, AF as a safety event, and AF from electrocardiogram (ECG). As part of the baseline interview, participants were asked about prior diagnosis of specific conditions, including AF. Self-reported AF was confirmed if participants answered affirmatively to the question: “Have you ever been told by a physician that you have: Atrial Fibrillation/Atrial Flutter?”. As part of the safety protocol, participants were regularly monitored by the trial’s safety committee for adverse events that were considered to be fatal or pose significant harm or disability to the participant [8]; AF events were among these adverse events. Finally, ECG AF was determined from 12-lead ECGs that indicated the presence of AF or atrial flutter. ECGs were performed at years 2 and 4, close-out visits, and any additional ECGs conducted for assessment of safety events and possible cardiovascular events [9]. ECG data were obtained at a 10 mm/mV calibration and speed of 25 mm/s using a GE MAC 1200 electrocardiograph (GE, Milwaukee, WI, USA). The ECG protocol has been published elsewhere [12]. Date of AF was the enrollment date for those with prevalent AF or the date of earliest evidence of AF for those with incident disease.

### 2.4. Ascertainment of Mild Cognitive Impairment and Probable Dementia

Cognitive status assessment protocols have been published elsewhere [13]. Briefly, participants underwent in-person assessments that included cognitive testing, or they were contacted for telephone interviews if they could not come to the trial center. Testing measures included the Montreal Cognitive Assessment (scores from 0 to 30), logical memory forms I (scores from 0 to 28) and II (scores from 0 to 14) of the Wechsler Memory Scale, and the digit symbol coding (scores from 0 to 135) test of the Wechsler Adult Intelligence Scale to assess processing speed [13]. If participants could not be contacted due to death or any other reason, trial examiners reached out to informants to administer the Dementia Questionnaire [14]. Cognitive status was confirmed through 2 adjudicators and categorized as MCI, PD, or no cognitive impairment. MCI was defined as at least 2 consecutive occurrences of adjudicated MCI.

### 2.5. Statistical Analysis

We examined the association of AF with MCI, PD, and a composite outcome of MCI and/or PD using Cox proportional hazard models. Model 1 was adjusted for age at randomization, sex, race/ethnicity, and education. Model 2 expands on model 1 by also accounting for OAC use, smoking status, history of CVD, BMI, total and HDL cholesterol, eGFR, and systolic and diastolic pressure. The proportional hazards assumption was tested for all outcomes using log-time interaction terms. Age for MCI (*p* = 0.03) and race/ethnicity for the composite MCI/PD outcome (*p* = 0.046) returned minor violations, which were addressed by adding an age-log time interaction term to the model when assessing MCI outcomes and stratifying by race/ethnicity when evaluating the composite outcome. AF was treated as a time-dependent exposure. AF status was divided into three categories for between-group comparisons: ECG AF (evidence of AF in any of the study ECGs), safety event AF only or safety event AF plus self-reported AF (without study ECG evidence of AF), and self-reported AF only. Self-report and awareness of AF symptoms may be tied to educational attainment; thus, we conducted an education-stratified analysis (college/no college) to assess the consistency of our results across strata. We also compared outcomes between sex (male/female) and race/ethnicity (White/Black) to assess potential demographic-based differences. All analyses were conducted using SAS 9.4 (Cary, NC; SAS Institute Inc, Cary, NC, USA).

## 3. Results

Of the 8469 participants included in the study (mean age at randomization: 67.8 ± 9.3), 35.0% were female and 58.4% were non-Hispanic White. Table 1 shows participant characteristics by AF status. Those with AF were older, less likely to be female, and more likely to be non-Hispanic White. Participants on average were overweight, with an average BMI of around 29. Over the course of the study, 974 participants either had AF at baseline or developed AF over the follow-up. 

Table 2 provides the breakdown of AF ascertainments by source. Overall, 310 participants had AF identified from study ECGs (32%), 221 from safety events but not ECGs (23%), and 443 exclusively from self-report (45%).

No differences were detected in the risk of MCI and/or PD when comparing those with any-source AF to those without AF (Table 3). After breaking down AF status by source diagnosis, those diagnosed by ECG had a 59% higher risk of MCI/PD (HR 1.59, 95%CI 1.06, 2.38) after adjustment for model 2 covariates (Table 4). AF diagnosed through safety events was not associated with any of the outcomes. Self-reported AF was associated with lower risk of MCI/PD (HR, 0.75; 95%CI, 0.55, 1.01). After stratifying the cohort by education, among those with no college experience, ECG AF was associated with 2 times the risk of MCI/PD (HR 2.12, 95%CI 1.19, 3.78), whereas no association was found in those with at least some college experience (Appendix A). Further examination, however, found no statistically significant interaction between education and ECG AF (*p*-value for interaction: 0.24). The association of self-reported AF or AF from safety events with MCI/PD did not differ by education. AF diagnosed through ECGs was associated with 77% higher risk of MCI/PD in men (HR 1.77, 95%CI 1.09, 2.88), but no such associations were found in women. That said, there was no evidence of interaction between sex and ECG AF (*p*-value for interaction: 0.38) (Appendix A). No statistically significant associations were found across race/ethnicity for any AF diagnostic method (Appendix A). 

## 4. Discussion

Within this cohort of individuals enrolled in SPRINT, we found no association of AF identified from a combination of three ascertainment sources with cognitive status among patients with hypertension. When comparing diagnoses by source, ECG-detected AF was associated with elevated risk of MCI and/or PD, while self-reported AF or AF identified from safety events (without study ECG confirmation) was not associated with MCI or PD risk. These results emphasize the importance of case definition in AF epidemiology. Since those with AF in a study ECG are more likely to have higher burden of AF, our findings indirectly highlight how AF burden may be related to dementia and cognitive decline. 

Multiple studies have consistently demonstrated that AF increases the risk of incident dementia and hastens cognitive decline [15,16]. Most of those studies have been conducted in the general population, and information on the impact of AF on cognitive outcomes in individuals at high risk of cardiovascular disease is scant. Of note, SPRINT is a cohort at higher cardiovascular risk compared to the general population. The average CHA₂DS₂-VASc score in this group is 2, which meets the ACC/AHA/Heart Rhythm Society’s threshold for recommending oral anticoagulants to high-risk patients [17]. An average ASCVD 10-risk score of 25% further emphasizes the high-risk nature of this older, hypertensive cohort [18]. Our analysis including AF from all sources suggested that AF did not increase MCI/PD risk, potentially indicating that, in high-risk individuals, the impact of AF on cognition is diminished. 

The abbreviated follow-up of the study may have prevented capture of an adequate number of MCI/PD cases, thereby making it difficult to characterize the long-term impact of AF on mild cognitive impairment and dementia. Furthermore, the effect of AF on cognitive decline may be mitigated in this older age cohort (mean age 67 years), as has been shown for other cardiovascular risk factors for dementia, including elevated blood pressure [19]. The association of AF with the risk of MCI and dementia is likely stronger for vascular than for Alzheimer’s disease-type dementia. Thus, not having data on dementia subtype may be obscuring differential risk profiles within the cohort. 

Differences in the association of AF with MCI/PD based on source of AF ascertainment require a more nuanced interpretation of our findings. No prior studies have evaluated whether differences in detection methods for AF impact its association with cognitive outcomes. In SPRINT, we found that self-reported AF was not associated with increased risk of MCI/PD. A common concern of self-reported AF is the likelihood of misclassification due both to false positives (patients reporting an AF diagnosis when they do not have the arrhythmia) and false negatives (not reporting the diagnosis). The validity of self-reported AF has been of particular concern in evaluating stroke risk. In the REGARDS study, however, self-reported AF was not only strongly associated with stroke but was also comparable to ECG-detected AF in terms of stroke risk [20]. Still, the validity of self-reported AF has not been formally evaluated.

Similarly, AF identified from safety events was not associated with MCI/PD risk. Though the validity of this ascertainment source is likely higher than self-report, AF cases identified specifically through this mechanism could be a particular subset of all AF cases with an overall lower cognitive risk profile. Unfortunately, SPRINT did not collect information on AF subtypes that would allow for the evaluation of this hypothesis.

Collecting data on the type and severity of AF (e.g., paroxysmal, permanent, persistent) or on the presence of AF symptoms was not part of the SPRINT trial protocol. About one-third of AF patients do not report symptoms [21]. That said, symptomatic status may not necessarily indicate future outcomes. In the AFFIRM trial, presence or absence of symptoms was not associated with differences in risk of stroke or death after adjusting for prior CVD events [22]. In another study, however, asymptomatic AF had a higher risk of thromboembolism and death than symptomatic AF [23]. Asymptomatic patients oftentimes are found to have persistent or permanent AF [23,24], which may be due to the fact that asymptomatic status may make it more difficult to detect possible AF, thereby forcing patients to live with undiagnosed AF for longer periods. 

Only AF diagnosed through ECGs was associated with MCI/PD. As the sporadic nature of paroxysmal AF may make it more difficult to detect through a standard 10-second ECG, ECG-diagnosed AF may be more likely to reflect persistent or permanent AF. Patients with persistent and permanent AF are known to have higher cardiovascular risk profiles and worse outcomes than patients with paroxysmal AF [25,26], and therefore may be at increased risk of adverse cognitive outcomes. Additionally, patients with persistent and permanent AF may be exposed for longer periods to the adverse mechanisms linking AF and dementia, such as cerebral hypoperfusion [27]. In a hypothesis-generating, cross-sectional ARIC study, persistent, but not paroxysmal, AF was associated with lower cognitive functioning [28]. Thus, while AF burden may be associated with cognitive decline, it may also lack the associated clinical symptoms leading to detection. However, there is not enough direct evidence, longitudinal or otherwise, connecting AF burden to cognitive outcomes [29].

There are strengths to this study. The large sample size and extensive array of biomarkers, lifestyle factors, and cardiovascular status and history offer a clearer picture of each participant’s overall risk. The SPRINT trial also maintained a dedicated cognitive functioning ancillary study that provided adjudicated diagnoses of MCI and PD. Furthermore, the 2.5-year follow-up period allowed us to examine longitudinal associations. That said, due to the success of the trial intervention, the study was concluded earlier than anticipated, therefore limiting the accrual of events. Another limitation of this study is the lack of information on AF duration/burden, which preclude us from drawing conclusions on the impact of AF severity on cognitive decline. Moreover, cognitive assessments were performed at baseline, ensuring that, at that time, participants did not have MCI. During follow-up, some participants may have developed MCI before being diagnosed with AF. These participants were not excluded from the study since the follow-up time up to the MCI diagnosis has to be considered, with follow-up time stopping at that time. If the participant was diagnosed with MCI before AF, then they were treated as not having AF from baseline to MCI diagnosis, since AF was evaluated as a time-dependent exposure.

## 5. Conclusions

In this group of hypertensive patients at high-cardiovascular risk, ECG-determined AF, but not AF ascertained from other sources, was associated with elevated risk of MCI and PD. These results highlight the need to incorporate measures of AF burden, duration and severity into future studies of AF and cognitive decline.

## Figures and Tables

**Figure 1 jcm-11-05800-f001:**
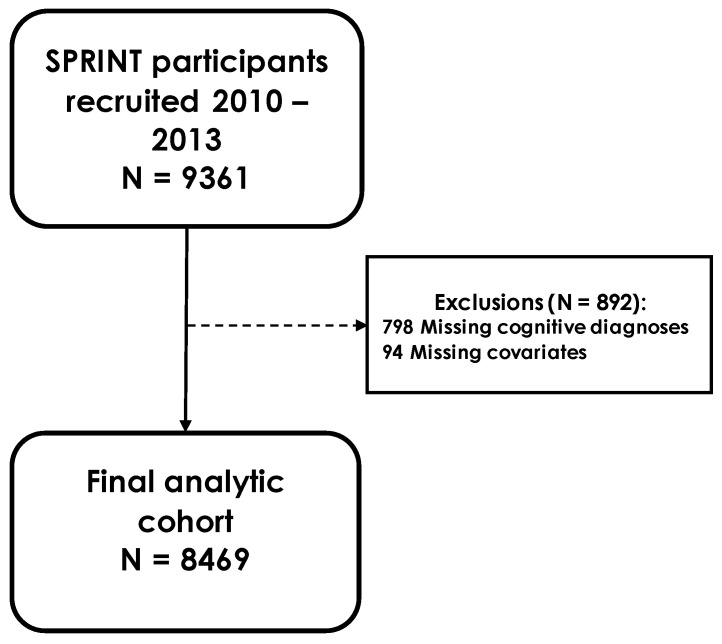
SPRINT participant flow chart.

**Table 1 jcm-11-05800-t001:** Participant characteristics by AF status, SPRINT.

	No AF (N = 7495)	AF (N = 974)
Age, years, mean (SD)	67.2 (9.2)	72.4 (9.0)
Female, %	35.8	29.4
Education, %		
Less than high school	8.1	6.9
High school and/or vocational school	24.4	22.5
At least some college	67.6	70.6
Race-ethnicity, %		
Non-Hispanic White	56.1	76.8
Non-Hispanic Black	31.2	15.9
Hispanic	11.0	5.3
Other	1.8	2.0
BMI (kg/m^2^)	29.8 (5.6)	29.6 (5.4)
Systolic blood pressure (mmHg), mean (SD)	139.6 (15.4)	139.6 (16.2)
Diastolic blood pressure (mmHg), mean (SD)	78.5 (11.7)	74.9 (12.2)
History of CVD ^a^, %	14.7	36.3
On oral anticoagulants, %	1.2	20.9
eGFR (mL/min per 1.73 m^2^), mean (SD)	72.5 (20.4)	67.3 (20.1)
HDL cholesterol (mg/dL), mean (SD)	52.7 (14.4)	53.0 (14.3)
Total cholesterol (mg/dL), mean (SD)	191.5 (41.1)	178.9 (39.5)
Currently smoking, %	13.5	8.1
CHA₂DS₂-VASc score ^b^	2.3 (1.1)	2.9 (1.1)
ASCVD 10-year risk score ^c^	21%	28%
MOCA score	23.0 (4.0)	23.1 (3.9)
Logical memory: immediate recall score	19.3 (4.8)	19.2 (4.9)
Logical memory: delayed recall score	8.3 (3.3)	8.1 (3.4)
Digit symbol coding score	51.4 (15.3)	49.5 (14.2)
Cognitive Outcomes		
Probable dementia, %	3.5	5.2
Mild cognitive impairment, %	7.4	8.5

AF, atrial fibrillation. BMI, body-mass index. CVD, cardiovascular disease. eGFR, estimated glomerular filtration rate. HDL, high-density lipoprotein cholesterol. ASCVD, atherosclerotic cardiovascular disease. MOCA, Montreal Cognitive Assessment. ^a^ Defined by self-report of angina, heart attack, or congestive heart failure; not including stroke or atrial fibrillation. ^b^ CHA₂DS₂-VASc variables include: congestive heart failure, age ≥ 75, stroke, vascular disease (myocardial infarction and peripheral vascular disease), age 65–74, female sex. All participants had hypertension, and no participant had diabetes as that was part of the exclusion criteria. ^c^ The ASCVDscore is based on the race- and sex-specific 2013 American College of Cardiology/American Heart Association pooled cohort equations. Coefficients for non-Hispanic Whites were used to calculate scores for Hispanic and other racial/ethnic groups.

**Table 2 jcm-11-05800-t002:** Breakdown of atrial fibrillation diagnostic sources, SPRINT.

Combinations of Diagnostic Sources			Classification in Analysis	N
Self-Report	Safety Event	ECG		
			Self-report	443
			Safety event	149
			ECG	97
			Safety event	72
			ECG	118
			ECG	57
			ECG	38
TOTAL:				974

ECG, electrocardiogram. The meaning of the background color is: diagnostic source of the Ns

**Table 3 jcm-11-05800-t003:** Hazard ratios and 95% confidence intervals for association of time-dependent AF (any diagnosis source) with probable dementia and MCI, SPRINT.

	No AF (N = 7495)	AF (N = 974)
Probable Dementia		
N. cases	265	51
Person years	36,093	4491
Incidence rate *	7.3	11.4
Model 1	1 (Ref.)	1.08 (0.78, 1.50)
Model 2	1 (Ref.)	1.04 (0.73, 1.48)
MCI		
N. cases	551	83
Person years	34,410	4223
Incidence rate	16.0	19.7
Model 1	1 (Ref.)	1.06 (0.82, 1.38)
Model 2	1 (Ref.)	1.14 (0.86, 1.50)
Probable dementia or MCI		
N. cases	738	120
Person years	34,716	4290
Incidence rate	21.3	28.0
Model 1	1 (Ref.)	0.96 (0.77, 1.19)
Model 2	1 (Ref.)	0.96 (0.76, 1.22)

AF, atrial fibrillation. MCI, mild cognitive impairment. * Crude incidence rate, per 1000 person years. Model 1: adjusted for age at randomization, sex, race/ethnicity and education. Model 2: adjusted for age at randomization, sex, race/ethnicity, education, OAC use, smoking, history of CVD, total cholesterol, HDL cholesterol, eGFR, systolic and diastolic pressure, and BMI.

**Table 4 jcm-11-05800-t004:** Hazard ratios and 95% confidence intervals for association of 4 levels of time-dependent AF status with probable dementia and MCI, SPRINT.

	No AF (N = 7495)	Self-Report Only AF (N = 443)	Safety Event AF Only or Self-Report + Safety Events (N = 221)	ECG AF (N = 310)
Probable Dementia				
N. cases	265	18	12	21
Person years	36,093	2108	1001	1383
Incidence rate *	7.3	8.5	12.0	15.2
Model 1	1 (Ref.)	0.88 (0.58, 1.34)	1.19 (0.68, 2.08)	1.40 (0.79, 2.48)
Model 2	1 (Ref.)	0.88 (0.57, 1.37)	1.10 (0.63, 1.92)	1.41 (0.78, 2.54)
MCI				
N. cases	551	26	20	37
Person years	34,410	2007	939	1278
Incidence rate	16.0	13.0	21.3	29.0
Model 1	1 (Ref.)	0.85 (0.61, 1.19)	1.03 (0.62, 1.71)	1.52 (0.96, 2.40)
Model 2	1 (Ref.)	0.91 (0.64, 1.30)	1.04 (0.62, 1.74)	1.53 (0.95, 2.47)
Prob dementia or MCI				
N. cases	738	40	28	52
Person years	34,716	2035	953	1303
Incidence rate	21.3	19.7	29.4	39.9
Model 1	1 (Ref.)	0.74 (0.56, 0.97)	0.97 (0.63, 1.48)	1.60 (1.08, 2.37)
Model 2	1 (Ref.)	0.75 (0.55, 1.01)	0.94 (0.62, 1.45)	1.59 (1.06, 2.38)

AF, atrial fibrillation. ECG, electrocardiogram. MCI, mild cognitive impairment. * Crude incidence rate, per 1000 p years. Model 1: adjusted for age at randomization, sex, race/ethnicity and education. Model 2: adjusted for age at randomization, sex, race/ethnicity, education, OAC use, smoking, history of CVD, total cholesterol, HDL cholesterol, eGFR, systolic and diastolic pressure, and BMI.

## Data Availability

Data used in this present study may be available to prospective investigators after providing proof of IRB/Ethics approval or of its exemption. Deidentified participant data and data dictionaries can be found in BioLINCC: https://biolincc.nhlbi.nih.gov/studies/sprint/.

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
