# Peer review of "Associations of Atrial Fibrillation with Mild Cognitive Impairment and Dementia: An Investigation Using SPRINT Research Materials"

_jcm, 2022, doi:10.3390/jcm11195800_

Round 1
Reviewer 1 Report
Atrial fibrillation (AF) is linked to increased risk of dementia and cognitive decline, but whether AF and its ascertainment methods affect cognition in patients with hypertension has been less often explored. Here the authors studied 8,469 participants with elevated systolic blood pressure who were free of stroke and diabetes at baseline enrolled in the Systolic Blood Pressure Intervention Trial. AF was ascertained using three approaches: self-report of AF, AF from a safety event, and study electrocardiogram-determined (ECG) AF. Mild cognitive impairment (MCI) and probable dementia (PD) were based off of in-person assessments or telephone interviews from the participant or an informant. They used Cox proportional hazard models to estimate hazard ratios for the association of AF (all 3 sources) with outcomes of MCI, PD, and a composite MCI/PD outcome. During a mean follow-up of 4.6 years, 974 (12%) participants had AF (prevalent or incident), 634 were diagnosed with MCI, and 316 with PD. When comparing those with AF (from any source) to those without, no differences were detected in the risk of MCI or PD. Comparison between AF sources found ECG-AF to be associated with an elevated risk of MCI and/or PD (hazard ratio [HR] 1.59, 95% confidence interval [95%CI] 1.06, 2.38). Neither AF ascertained through safety 25 events nor self-reported AF were associated with MCI or PD. The association of AF with incidence of MCI/PD differed by method of AF ascertainment. Case definition of AF and quantification of AF burden are important factors in studies evaluating the link between AF and cognitive dysfunction. There are a number of issues for this work.
1. How can we exclude the patients with onset of MCI prior to development of AF.
2. The work is none-original and it would be wise to include more studies for meta-analysis.
3. Cognitive score available?
Author Response
How can we exclude the patients with onset of MCI prior to development of AF.
Response:
- Cognitive assessments were performed at baseline, and at that time participants did not have MCI. During follow-up, some participants may have developed MCI before being diagnosed with AF, but they were not excluded from the study since the follow-up time up to the MCI diagnosis has to be included, with follow-up time stopping at that time. If the participant was diagnosed with MCI before AF, then they were simply treated as not having AF from baseline to MCI diagnosis.
The work is none-original and it would be wise to include more studies for meta-analysis.
Response:
- We acknowledge that prior studies have explored the association of AF with risk of dementia, and some meta-analyses on this topic have been published (e.g. PMID 31780919, 34244959, 34861843). Our goal in this analysis was to evaluate this question in a specific population (hypertensive, high-risk cardiovascular patients) and to explore the impact that different AF ascertainment sources had on our results. Conducting a systematic review and meta-analysis goes beyond the scope of this manuscript. Our results support previous work in the study of the association between AF and MCI/dementia and highlight the importance of AF case definition in studies concerning cognitive decline.
Cognitive score available?
Response:
- We agree that cognitive scores of participants at baseline would be insightful information. We have included baseline scores from tests that were later used to determine cognitive status to Table 1 along with appropriate footnotes. These tests include the Montreal Cognitive Assessment (MOCA), logical memory subtests from the Wechsler Memory Scale, and the digit symbol coding test from the Wechsler Adult Intelligence Scale used to test processing speed. We have also included some more clarification in the text at lines 104-107:
- “Testing measures included the Montreal Cognitive Assessment (scores from 0 to 30), logi-cal memory subtests forms I (scores from 0 to 28) and II (scores from 0 to 14) of the Wechsler Memory Scale, and the digit symbol coding (scores from 0 to 135) test of the Wechsler Adult Intelligence Scale to assess processing speed.”
We thank you for your feedback.
Reviewer 2 Report
Dear Sir/Madam,
I had the opportunity to act as a reviewer on the recent submission by Alam et al. to the Journal of Clinical Medicine.
The authors present original research studying the association of atrial fibrillation with mild cognitive impairment and dementia in the SPRINT study cohort. The authors found that the association of atrial fibrillation with incidence of mild cognitive impairment/probable dementia differ by method of ascertainment of atrial fibrillation.
The manuscript is well structured; however, some issues need to be addressed:
- The cohort is not good enough characterized: the baseline characteristics should include at least the SCORE score and CHA2DS2VASc score (as a virtual score for patients without atrial fibrillation).
- The authors aim at studying associations of atrial fibrillation with mild cognitive impairment and probable dementia over a period of 3 years and the association of atrial fibrillation with mild cognitive impairment and probable dementia by method of atrial fibrillation ascertainment. However, the authors comment in the discussion only on the results of the second aim.
Best regards,
Author Response
I had the opportunity to act as a reviewer on the recent submission by Alam et al. to the Journal of Clinical Medicine.
The authors present original research studying the association of atrial fibrillation with mild cognitive impairment and dementia in the SPRINT study cohort. The authors found that the association of atrial fibrillation with incidence of mild cognitive impairment/probable dementia differ by method of ascertainment of atrial fibrillation.
The manuscript is well structured; however, some issues need to be addressed:
The cohort is not good enough characterized: the baseline characteristics should include at least the SCORE score and CHA2DS2VASc score (as a virtual score for patients without atrial fibrillation).
Response:
- We agree that more information is needed to sufficiently understand the underlying population of this study. Per your recommendations, we have added CHAâ‚‚DSâ‚‚-VASc scores to Table 1, along with ASCVD 10-year risk scores generated by the 2013 ACC/AHA pooled cohort equations, which were developed for the US population.
The authors aim at studying associations of atrial fibrillation with mild cognitive impairment and probable dementia over a period of 3 years and the association of atrial fibrillation with mild cognitive impairment and probable dementia by method of atrial fibrillation ascertainment. However, the authors comment in the discussion only on the results of the second aim.
Response:
- Thank you for bringing that to our attention. We have expanded our discussion section to address our first aim, which focused on the association of AF (from all 3 sources) with MCI/PD, in more depth:
- “The abbreviated follow-up of the study may have prevented capture of an adequate number of MCI/PD cases, thereby making it difficult to characterize the long-term impact of AF on mild cognitive impairment and dementia. Furthermore, the effect of AF on cognitive decline may be mitigated in this older age cohort (mean age 67 years), as has been shown for other cardiovascular risk factors for dementia, including elevated blood pressure. The association of AF with the risk of MCI and dementia is likely stronger for vascular than for Alzheimer’s disease-type dementia. Thus, not having data on dementia subtype may be obscuring differential risk profiles within the cohort.
- Differences in the association of AF with MCI/PD based on source of AF ascertainment require a more nuanced interpretation of our findings. No prior studies have evaluated whether differences in detection methods for AF impact its association with cognitive outcomes.” (lines 198-209)
We thank you for your feedback.
Round 2
Reviewer 1 Report
The authors have addressed my concerns in a fine manner
Author Response
"The authors have addressed my concerns in a fine manner"
Response: We are glad to have been able to address your concerns with our manuscript. Thank you for your time.
Reviewer 2 Report
Dear Sir/Madam,
Thank you for reviewing the manuscript and addressing the mentioned issues. These were adequately answered. Therefore, the manuscript seems suitable for publishing in the present form.
Best regards
Author Response
“Thank you for reviewing the manuscript and addressing the mentioned issues. These were adequately answered. Therefore, the manuscript seems suitable for publishing in the present form.”
Response: We are glad to have been able to address your concerns with our manuscript. Thank you for your time.